# Predictors of Willingness of the General Public to Receive a Second COVID-19 Booster Dose or a New COVID-19 Vaccine: A Cross-Sectional Study in Greece

**DOI:** 10.3390/vaccines10071061

**Published:** 2022-06-30

**Authors:** Petros Galanis, Irene Vraka, Aglaia Katsiroumpa, Olga Siskou, Olympia Konstantakopoulou, Theodoros Katsoulas, Theodoros Mariolis-Sapsakos, Daphne Kaitelidou

**Affiliations:** 1Clinical and Epidemiology Laboratory, Faculty of Nursing, National and Kapodistrian University of Athens, 11527 Athens, Greece; aglaiakat@nurs.uoa.gr; 2Department of Radiology, P. & A. Kyriakou Children’s Hospital, 11527 Athens, Greece; irenevraka@yahoo.gr; 3Department of Tourism Studies, University of Piraeus, 18534 Piraeus, Greece; olsiskou@nurs.uoa.gr; 4Center for Health Services Management and Evaluation, Faculty of Nursing, National and Kapodistrian University of Athens, 11527 Athens, Greece; olykonstant@nurs.uoa.gr (O.K.); dkaitelid@nurs.uoa.gr (D.K.); 5Faculty of Nursing, National and Kapodistrian University of Athens, 11527 Athens, Greece; tkatsoul@nurs.uoa.gr (T.K.); tmariolis@nurs.uoa.gr (T.M.-S.)

**Keywords:** second COVID-19 booster, vaccination, COVID-19, willingness, predictors, general population

## Abstract

Given the concerns of waning immunity from the primary COVID-19 vaccines and the first booster dose, we conducted an online cross-sectional study in May 2022 to investigate willingness to receive a second COVID-19 booster dose or a new COVID-19 vaccine and its associated factors. Overall, 62% of the participants were willing to be vaccinated, 25.8% were unsure, and 12.3% were unwilling to be vaccinated. The main reasons against accepting a second COVID-19 booster dose/new COVID-19 vaccine were concerns about the side effects and the effectiveness and the opinion that further vaccination is unnecessary. Males, younger individuals, participants without a previous COVID-19 diagnosis, and those with good/very good self-perceived physical health were significantly more frequently willing to receive a second COVID-19 booster dose or a new COVID-19 vaccine. Additionally, increased fear of the COVID-19, increased trust in COVID-19 vaccinations, and decreased fear of a second booster dose or a new COVID-19 vaccine was associated with increased willingness. Our results show some hesitancy and unwillingness toward further COVID-19 vaccination and indicate that the fear of COVID-19 and trust in COVID-19 vaccination affects public opinion.

## 1. Introduction

New variants of SARS-CoV-2 have driven several countries to adopt a COVID-19 vaccine booster. Early evidence supports the efficacy of the booster dose against the new variants (Delta and Omicron) [1,2]. On the other hand, the existing booster vaccination program produces broad but incomplete immunity against SARS-CoV-2 variants, including Omicron [3]. Moreover, real-world evidence has shown waning first booster dose effectiveness over time against new COVID-19 infections and hospitalization, especially for the Omicron variant [4].

A second booster dose could help increase protection levels, especially for individuals in high-risk groups. Thus, different countries have already recommended a fourth COVID-19 mRNA vaccine dose (second booster) for older and immunocompromised individuals. For instance, the U.S. Food and Drug Administration authorized a second booster dose for certain individuals, considering the known and potential benefits and risks [5]. A second booster dose of a COVID-19 mRNA vaccine is effective in reducing the rates of SARS-CoV-2 infection, severe COVID-19, the short-term risk of COVID-19-related outcomes, and hospitalization and mortality due to COVID-19 [6,7,8,9,10]. However, first COVID-19 booster uptake is low and even lower for the second dose. For instance, as of May 2022, 46.6% of individuals in the USA who are fully vaccinated opted to receive the first booster [11]. Moreover, in that population, 69.5% of individuals over 65 years of age received the first booster dose, while the corresponding percentage for the second booster dose was 26.7%.

Thus, individuals’ willingness to accept a first booster dose is an important area of research. Several studies investigated the willingness of the general public to receive a first COVID-19 booster dose [12,13,14,15,16,17,18,19,20,21,22,23,24]. Among these studies, the percentage of individuals who were willing to take a first booster dose ranges from 44.6% [12] to 95.5% [14], while the percentage of unwilling individuals ranges from 1.0% [22] to 30.7% [12].

The literature suggests that older age, influenza vaccination, and confidence in COVID-19 vaccines are associated with the willingness of the general public to accept a booster, while adverse side effects after previous COVID-19 vaccination and concerns for serious side effects after booster doses are associated with vaccine hesitancy [12,13,14,15,16,17,18,19,20,21,22,23,24].

However, the intention of the public to receive a second booster dose remains unknown. Moreover, a second booster or even a new COVID-19 vaccine may be very important in the fall of 2022, since the emergence of new COVID-19 viral variants is always possible. Therefore, the purpose of this study was to investigate the willingness of the general public to receive a second COVID-19 booster dose or a new COVID-19 vaccine and its associated factors.

## 2. Materials and Methods

### 2.1. Study Design and Participants

On 5 April 2022, the Greek Committee on Vaccination announced that a second booster dose was recommended only to vulnerable groups that had been given their first booster dose at least four months previously [25]. Vulnerable groups included adults over 70 years of age, adults over 60 years of age suffering from at least one chronic condition, and the elderly in nursing or care homes. We performed this cross-sectional study from 23 May to 30 May 2022 using a convenience sample. Until the time of our study, a second booster is not being recommended to the general population. To be eligible for inclusion, participants had to be aged 18 years or above, had to understand the Greek language, and have completed a COVID-19 vaccine course (i.e., first COVID-19 doses and a booster dose). We created an anonymous version of the study questionnaire with Google forms and then disseminated it through social media platforms and personal e-mails. We included a cover letter in the online self-administered questionnaire to inform the participants that data are anonymous and participation in the study is voluntary. The participants filled in their e-mail to allow us to later remove those who had participated in the study more than once. The study flowchart is presented in Figure 1. The principles of the Declaration of Helsinki were applied in our study and the Ethics Committee of Department of Nursing, National and Kapodistrian University of Athens, approved the protocol (reference number; 370, 02-09-2021).

### 2.2. Predictor Variables

Predictor variables included socio-demographic variables, COVID-19 related variables, and attitudes toward the COVID-19 vaccination and pandemic. Socio-demographic variables included gender (female or male), age (continuous variable), marital status (single, married, in a couple relationship without marriage, divorced, or widowed), educational level (elementary school, high school), chronic disease (no or yes), self-assessment of physical health (very poor, poor, average, good, or very good), and influenza vaccination during 2021 (no or yes).

COVID-19 related variables included previous COVID-19 diagnosis (no or yes), hospitalization due to COVID-19 (no or yes), COVID-19-related death in family members/friends (no or yes), and adverse reactions and discomfort experienced after previous COVID-19 vaccine doses (scale from 0 (none) to 10 (great discomfort)).

Additionally, we used a valid questionnaire in Greek to measure attitudes toward COVID-19 vaccination and the pandemic [26]. There are four factors in this questionnaire: fear of COVID-19; information regarding the COVID-19 pandemic and vaccination; compliance with hygiene measures; and trust in COVID-19 vaccination. We calculated Cronbach’s alpha for these four factors (0.85, 0.81, 0.71, and 0.78 respectively) and we found very good reliability of the questionnaire. Responses ranged from 0 (totally disagree) to 10 (totally agree). Additionally, a total score from 0 to 10 was calculated for each factor. Higher values indicate higher level of fear, information, compliance, and trust.

Moreover, we used three developed study items to measure the attitudes of participants toward a second booster dose/new COVID-19 vaccine: I am afraid to have a second booster dose/new COVID-19 vaccine; I have concerns about the long-term side effects of a second booster dose/new COVID-19 vaccine; I feel protected by the previous COVID-19 vaccine doses. We measured these items in a scale ranged from 0 (totally disagree) to 10 (totally agree).

### 2.3. Outcome Variable

Willingness to receive a second booster dose or a new COVID-19 vaccine was measured with a question (“If a second booster dose or a new COVID-19 vaccine is recommended as a supplement to the current vaccination schedule, would you accept it?”). Response options were the following: definitely no, probably no, unsure, probably yes, and definitely yes. We decided to investigate the willingness both for a second booster and a new COVID-19 vaccine since pharmaceutical companies are now seeking new safe and effective COVID-19 vaccines that will provide a longer duration of protection and increase immunity [27]. Thus, a new COVID-19 vaccine in the fall of 2022 could be as important as a second booster or even more important.

Additionally, we measured the reasons participants provided for potentially refusing a second COVID-19 booster dose or a new COVID-19 vaccine with a single item (“Which of the following concerns best describe why you might refuse to accept a second COVID-19 booster dose or a new COVID-19 vaccine?”). Available answers were the following: I have doubts about the COVID-19 vaccines safety; I have doubts about the COVID-19 vaccines effectiveness; I worry about the short-term side effects; I have a low risk of infection; I am healthy and I have a low risk of COVID-19-related complications; I am not convinced that another dose will be necessary; I do need it because I believe I have immunity against the SARS-CoV-2; I have already been diagnosed with COVID-19, so I think another booster dose would not be beneficial; I am tired of the vaccination process; I worry about the long-term side effects; Other concerns (please specify).

### 2.4. Statistical Analysis

We use absolute (n) and relative (%) frequencies to present categorical variables. Continuous variables are presented as mean and standard deviation. Our aim was to investigate predictors of public willingness to receive a second COVID-19 booster dose or a new COVID-19 vaccine. Thus, we coded the outcome variable so that those who answered “probably yes” or “definitely yes” (willing participants to receive a second booster dose or a new COVID-19 vaccine) were compared to those who answered “definitely no”, “probably no” or “unsure” (unwilling or hesitant participants to receive a second booster dose or a new COVID-19 vaccine). We performed univariate and multivariable logistic regression analysis to examine the associations between the predictor variables and willingness to receive a second booster dose or a new COVID-19 vaccine. We included all independent variables in a multivariable logistic regression model in order to eliminate confounding. We calculated unadjusted and adjusted odds ratios (aOR) with corresponding 95% confidence intervals (CI) and *p*-values. A *p*-value of < 0.05 was considered statistically significant. Statistical analysis was performed with the Statistical Package for Social Sciences software (IBM Corp. Released 2012. IBM SPSS Statistics for Windows, Version 21.0. Armonk, NY, USA: IBM Corp.).

## 3. Results

### 3.1. Socio-Demographic Characteristics 

The study population included 815 participants with a mean age of 37 years. Among the participants, 76.1% were females, 82.4% indicated that their physical health was good/very good, 33.1% received an influenza vaccine during the previous season, and 23.3% suffered from a chronic condition. We present the socio-demographic characteristics of the study population in Table 1.

### 3.2. Willingness to Receive a Second Booster Dose or a New COVID-19 Vaccine

As for the willingness to receive a second booster dose or a new COVID-19 vaccine, 22.7% of participants in our study were willing to be vaccinated; 39.3% were unsure but tended towards willing; 25.7% were unsure; 4.9% were unsure but tended towards unwilling; and 7.4% were unwilling to be vaccinated. Thus, 62% of the participants were willing to receive a second booster dose or a new COVID-19 vaccine, while 38% were unwilling or hesitant. The main reasons to hesitate or to be unsure over the second booster dose or a new COVID-19 vaccine were as follows: “I worry about the long-term side effects” (46.8%); “I am not convinced that another dose will be necessary” (40.3%); “I have doubts about the COVID-19 vaccines effectiveness” (30.7%); “I worry about the short-term side effects” (29%); “I am tired of the vaccination process” (27.4%); “I am healthy and I have a low risk of COVID-19-related complications” (22.6%); and “I do not need it because I believe I have immunity against the SARS-CoV-2” (19.4%). Participants experienced a low level of adverse reactions and discomfort after previous COVID-19 vaccine doses. Table 2 depicts the participants’ willingness to receive a second booster dose or a new COVID-19 vaccine.

### 3.3. COVID-19 Related Variables and Attitudes toward COVID-19 Vaccination and Pandemic

Nearly half of the participants (50.9%) were previously diagnosed with COVID-19 and among them 3.6% were hospitalized. Moreover, 31.3% of the participants had family members/friends who had died because of COVID-19. Participants’ fear of COVID-19, second booster dose/new COVID-19 vaccines, and long-term adverse effects of COVID-19 vaccination was moderate. Additionally, we recorded a moderate to high level of participants’ confidence in COVID-19 vaccination. In addition, participants stated that they are well informed about the COVID-19 pandemic and that they are highly compliant with hygiene measures. COVID-19-related variables and the attitudes of the participants toward COVID-19 vaccination and the pandemic are summarized in Table 2.

### 3.4. Predictors of Willingness

Results from the multivariable logistic regression model predicting the willingness of the participants to receive a second booster dose or a new COVID-19 vaccine from socio-demographic factors, COVID-19 related variables, and attitudes toward COVID-19 vaccination and pandemic are shown in Table 3. We found that high levels of both fear of COVID-19 (aOR = 1.73, 95% CI: 1.47–2.03) and trust in the COVID-19 vaccination (aOR = 2.11, 95% CI: 1.69–2.63) were associated with willingness. In addition, low levels of fear of a second booster dose or a new COVID-19 vaccine was associated with willingness (aOR = 0.66, 95% CI: 0.59–0.75). Additionally, participants who were not previously diagnosed with COVID-19 (aOR = 2.96, 95% CI: 1.84–4.75) were more likely to be willing to receive a second booster dose or a new COVID-19 vaccine. Furthermore, participants with good/very good self-perceived physical health were nearly four times more likely than participants with very poor/poor/moderate self-perceived physical health to be willing to receive a second booster dose or a new COVID-19 vaccine (aOR = 3.63, 95% CI: 1.78–7.42). Among socio-demographic factors, gender was a significant predictor of willingness. Being male increased the likelihood of a second booster dose or new COVID-19 vaccine acceptability by nearly 2.5 times (aOR = 2.40, 95% CI: 1.34–4.29). Moreover, decreased age was associated with increased willingness (aOR = 0.97, 95% CI: 0.95–0.99). Interestingly, the impact of age was the opposite in univariate logistic regression analysis (OR = 1.03, 95% CI: 1.02–1.04), while the impact of the other independent variables remained stable.

## 4. Discussion

To the best of our knowledge, this is the first study that investigated public willingness of acceptance of a second COVID-19 booster dose/new COVID-19 vaccine. Moreover, we identified predictors of this willingness, including socio-demographic variables, COVID-19 related variables, and attitudes toward COVID-19 vaccination and the pandemic. Thus, we compared our results with studies that investigated public intention to accept the first COVID-19 booster dose.

In our study, 62% of the participants were willing to accept a second COVID-19 booster dose or a new COVID-19 vaccine, while a significant percentage (12.3%) expressed reluctance to do so, and 25.7% were unsure. Our results suggest that public willingness to receive a second booster is lower than willingness to receive a first booster. In particular, median public willingness to receive a first booster was 81% [12,13,14,15,16,17,18,19,20,21,22,23,24]. Moreover, the inclination of the general population in European countries to accept a first booster is higher than the inclination of our sample to accept a second booster: 95.5% in Denmark [14], 92.3% in United Kingdom [18], and 71% in Poland [20]. In addition, the actual uptake rate of a second booster or a new COVID-19 vaccine may be even lower than the willingness rate in our study, since people who say they will probably have a second booster or a new COVID-19 vaccine may eventually decide otherwise. The lower acceptance rate of the second booster may be partly explained by higher public expectations of the effectiveness and the safety of the initial COVID-19 vaccine doses and the first booster. Perceived effectiveness of the COVID-19 vaccines is an important motivator but public perception of effectiveness does not always align with scientists’ and policy makers’ views [28]. For instance, the public may expect that an effective COVID-19 vaccination program would enable the elimination of the SARS-CoV-2 and warrant a return to normal. In addition, safety-related issues are still a key consideration in individual decision-making, especially for an additional booster dose or a new COVID-19 vaccine.

We found that the primary reasons for refusing a second booster/new COVID-19 vaccine were worries about safety, effectiveness, and side effects; the low self-perceived risk of COVID-19 complications; and the belief that initial COVID-19 vaccine doses provide sufficient protection against the disease. These findings are confirmed by studies investigating the refusal of individuals to accept both the first COVID-19 booster dose [15,16,20,23,24] and the initial COVID-19 vaccine doses [29,30,31,32]. It is reasonable that individuals with a complete COVID-19 vaccine course are concerned about the need, safety, and effectiveness of another booster or a new vaccine and policy makers should consider these issues when they design education, communication, and policy-based interventions about COVID-19 vaccines [29,31]. Public concerns about COVID-19 vaccine safety are still an obstacle for booster uptake. Thus, reliable information regarding the need for and importance of boosters should be provided to previously vaccinated individuals. For instance, valid information regarding post-marketing surveillance and compensation policy after side effects would be critical in our efforts to persuade people to accept booster doses [33,34]. Unfortunately, initial COVID-19 vaccine doses cannot provide sufficient immunization since the literature suggests that antibody levels and vaccine effectiveness decreases over time even after a first booster [3,4,35].

We found a positive association between confidence in COVID-19 vaccination and second booster acceptance. Previous studies confirm this finding since confidence in the safety and effectiveness of COVID-19 vaccines and trust in pharmaceutical companies improved first COVID-19 booster acceptance rate and increased the number of vaccinated people [15,16,20,21,36,37]. Moreover, vaccine efficacy and effectiveness is associated with COVID-19 vaccine acceptance and uptake [38,39,40,41]. Thus, increased vaccine efficacy and effectiveness could enhance confidence in COVID-19 vaccines, leading to more people being vaccinated. Public health education and intervention programs are important to improve confidence and reduce perceived safety barriers if a second booster or a new COVID-19 vaccine is approved for the general population in the future.

Our findings also indicated that the decreased fear of a second COVID-19 booster dose or a new COVID-19 vaccine was associated with increased odds of accepting vaccination. The recent literature suggests that adverse reactions and discomfort experienced after the initial COVID-19 vaccine doses are one of the most common causes of first booster rejection [12,16,17,20,22]. The fear of an additional booster dose is a reasonable feeling especially among individuals that experienced side effects after the initial doses or/and the first booster dose. Thus, policy makers should emphasize that COVID-19 vaccines and booster doses confer high levels of protection against COVID-19-related outcomes, such as hospitalization, complications, and death.

Our results showed that there was a positive association between the fear of COVID-19 and willingness to accept a second COVID-19 booster dose/new COVID-19 vaccine. The literature confirms this finding since higher self-perceived COVID-19 vulnerability is associated with COVID-19 vaccination uptake [42]. Additionally, a systematic review of studies with parents found an association between the perceived threat from the COVID-19 and parents’ decision to accept a COVID-19 vaccine for their children [43]. The COVID-19 pandemic has caused great fear, which then leads to mental health issues, i.e., anxiety, stress, depression, sleep problems, and lack of mental well-being [44,45]. Thus, there is a need to design and implement appropriate health promotion programs in order to reduce fear of COVID-19 and improve mental health.

Interestingly, previous COVID-19 diagnosis had an inverse association with willingness. Recent data suggest that individuals with a prior COVID-19 infection had lower odds of intention to accept a COVID-19 vaccine [46] and lower odds of vaccine uptake [47] than those without a prior COVID-19 infection. In addition, uncertainty about the acceptance of a COVID-19 vaccine is higher among individuals with a prior COVID-19 infection [48]. Individuals who had previously been infected with COVID-19 and who are fully vaccinated with initial doses and the first booster may believe that they have immunity against the SARS-CoV-2.

Regarding socio-demographic factors, we found that younger participants were more likely to accept a second COVID-19 booster dose or a new COVID-19 vaccine than older individuals. Age is a controversial issue since some studies found that increased age is associated with increased acceptance of a first booster dose [16,17,18,20,22,24], while other studies found the opposite [15,21]. Low acceptance of a second COVID-19 booster dose or a new COVID-19 vaccine among elderly persons would be of great concern since the elderly are a high-risk group for COVID-19-related complications and mortality. On the other hand, low acceptance among younger persons is also of great importance since they are most likely to spread SARS-CoV-2.

Our findings also indicated that males had higher odds of accepting a second COVID-19 booster dose or a new COVID-19 vaccine than females. Gender is also a complex issue since some studies found that males were more willing to accept a first booster [16,21], while other studies showed the opposite conclusion [17,20]. Low acceptance among females could be linked to psychological and hormonal gender differences [49,50].

### Limitations

We conducted a web-based survey that is easily susceptible to the effects of selection bias. In particular, elderly persons were underrepresented in our study as confirmed by the age distribution of the participants. The median age of our sample was 37 years, while median age in Greece is 45.6 years [51]. Additionally, almost 3/4 of our participants had a university level degree and 1/4 had completed high school. Therefore, an oversampling of highly educated individuals is apparent in our study. Moreover, we expect that participation rate was higher among persons who are interested in health issues, such as vaccination and the COVID-19 pandemic. Unfortunately, it was impossible to obtain data on non-participants in order to make valid comparisons with participants. Thus, we cannot confidently confirm our findings’ generalizability in other populations. In addition, we investigated several socio-demographic variables, COVID-19 related variables, and attitudes toward COVID-19 vaccination and the pandemic as predictors of public willingness to receive a second COVID-19 booster dose or a new COVID-19 vaccine. However, there is still room for researchers to explore other predictors, e.g., income, occupation, media preferences, sources of information, knowledge level, etc. Although vaccination intentions tend to strongly predict actual behavior, the actual uptake of a second COVID-19 booster dose or a new COVID-19 vaccine should be investigated in the future. Additionally, our findings reflect a snapshot of public willingness to accept a second COVID-19 booster dose or a new COVID-19 vaccine, but individual attitudes are dynamic and evolving. Thus, researchers should conduct longitudinal studies that measure changes in COVID-19 vaccination intentions over time. Self-reporting bias is always probable in studies with self-administrated questionnaires. In particular, willingness rates in our study may be an overestimation due to social desirability. Finally, we used an online questionnaire in Greek. Thus, only persons who understood the Greek language could participate in our study. In which case, the participation rate of minority groups, such as migrants, is expected to be very low.

## 5. Conclusions

Our findings present important insights related to the second COVID-19 booster dose or new COVID-19 vaccine willingness and potential factors related to booster/vaccine willingness. A significant proportion of those who have been vaccinated against COVID-19 with initial doses and a first booster dose reported the intention to accept a second COVID-19 booster dose or a new COVID-19 vaccine. This is very encouraging since booster shots or/and new COVID-19 vaccines will be essential in the control of the COVID-19 pandemic, especially if new COVID-19 viral variants emerge. However, our results show some hesitancy and unwillingness toward further COVID-19 vaccination among those who are fully vaccinated against COVID-19. It is an urgent need to find solutions to change this attitude. For instance, the identification of the potential predictors of COVID-19 vaccine willingness could prove essential in encouraging future uptake. Therefore, policy makers should develop public health interventions and effective communication strategies, emphasizing the safety and the effectiveness of boosters in order to increase the COVID-19 vaccination uptake. Since evidence shows that existing COVID-19 vaccines offer protection for limited time periods, there is an urgent need to investigate ongoing attitudes of individuals toward boosters and new vaccines.

## Figures and Tables

**Figure 1 vaccines-10-01061-f001:**
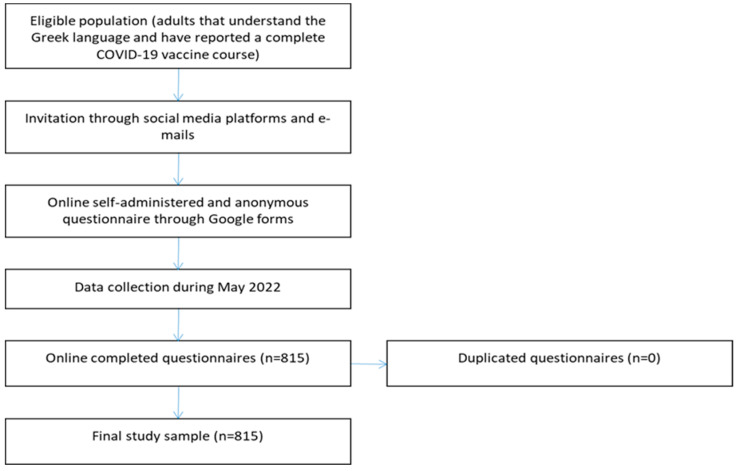
Study flowchart.

**Table 1 vaccines-10-01061-t001:** Sample socio-demographic characteristics (n = 815).

Characteristics	N	%
Gender		
Male	195	23.9
Female	620	76.1
Age (years) ^a^	37.0	13.3
Marital status		
Single	440	54.0
Married or in a relationship without marriage	290	35.6
Divorced	75	9.2
Widowed	10	1.2
Educational level		
Elementary school	0	0.0
High school	225	27.6
University degree	590	72.4
Chronic disease		
No	625	76.7
Yes	190	23.3
Self-perceived physical health		
Very poor	0	0.0
Poor	5	0.6
Moderate	90	11.0
Good	400	49.1
Very good	320	39.3
Influenza vaccination during 2021		
No	545	66.9
Yes	270	33.1

^a^ mean, standard deviation.

**Table 2 vaccines-10-01061-t002:** COVID-19 related variables, willingness of the participants to receive a second booster dose or a new COVID-19 vaccine, and attitudes toward COVID-19 vaccination and pandemic (n = 815).

	N	%
Previous COVID-19 diagnosis		
No	400	49.1
Yes	415	50.9
Hospitalization due to COVID-19 (n = 415)		
No	400	96.4
Yes	15	3.6
COVID-19-related death in family members/friends		
No	560	68.7
Yes	255	31.3
If a second booster dose or a new COVID-19 vaccine is recommended as a supplement to the current vaccination schedule, would you accept it?		
Definitely no	60	7.4
Probably no	40	4.9
Unsure	210	25.7
Probably yes	320	39.3
Definitely yes	185	22.7
Which of the following concerns best describe why you might refuse to accept a second COVID-19 booster dose or a new COVID-19 vaccine? (n = 310)		
I have doubts about the COVID-19 vaccine safety	40	12.9
I have doubts about the COVID-19 vaccine effectiveness	95	30.7
I worry about the short-term side effects	90	29.0
I have a low risk of infection	0	0.0
I am healthy and I am at low risk of COVID-19-related complications	70	22.6
I am not convinced that another dose will be necessary	125	40.3
I do not need it because I believe I have immunity against the SARS-CoV-2	60	19.4
I have already been diagnosed with COVID-19, so I think another booster dose would not be beneficial	65	21.0
I am tired of the vaccination process	85	27.4
I worry about the long-term side effects	145	46.8
Adverse reactions and discomfort experienced after previous COVID-19 vaccine doses	2.8	2.3
Fear of the COVID-19 ^a^	5.6	2.0
Information regarding the COVID-19 pandemic and vaccination ^a^	8.1	1.5
Compliance with hygiene measures ^a^	9.0	1.1
Trust in COVID-19 vaccination ^a^	7.0	1.6
Fear of a second booster dose or a new COVID-19 vaccine ^a^	3.6	3.3
Concerns about the long-term side effects of a second booster dose or a new COVID-19 vaccine ^a^	4.0	3.1
Self-perceived protection by the previous COVID-19 vaccine doses ^a^	7.0	2.3

^a^ mean, standard deviation.

**Table 3 vaccines-10-01061-t003:** Univariate and multivariable logistic regression analysis with the willingness of the participants to receive a COVID-19 vaccine second booster dose or a new COVID-19 vaccine as the dependent variable (reference: unwilling or hesitant participants).

Variable	Unadjusted OR (95% CI)	*p*-Value	Adjusted OR (95% CI) ^a^	*p*-Value
Gender (male vs. female)	2.48 (1.72–3.60)	<0.001	2.40 (1.34–4.29)	**0.003**
Age (years)	1.03 (1.02–1.04)	<0.001	0.97 (0.95–0.99)	**0.02**
Marital status (married vs. single/widowed/divorced)	2.05 (1.50–2.79)	<0.001	1.29 (0.75–2.24)	0.36
Educational level (university degree vs. high school)	1.28 (0.93–1.74)	0.13	1.00 (0.54–1.85)	0.99
Chronic disease (yes vs. no)	0.92 (0.66–1.29)	0.64	1.26 (0.74–2.14)	0.40
Self-perceived physical health (good/very good vs. very poor/poor/moderate)	2.51 (1.62–3.87)	<0.001	3.63 (1.78–7.42)	**<0.001**
Influenza vaccination during 2021 (yes vs. no)	2.58 (1.86–3.57)	<0.001	1.59 (0.95–2.66)	0.08
Previous COVID-19 diagnosis (no vs. yes)	3.07 (2.28–4.14)	<0.001	2.96 (1.84–4.75)	**<0.001**
COVID-19-related death in family members/friends (yes vs. no)	1.19 (0.87–1.61)	0.28	1.37 (0.82–2.29)	0.24
Adverse reactions and discomfort experienced after previous COVID-19 vaccine doses	0.86 (0.81–0.92)	<0.001	1.13 (0.99–1.28)	0.05
Fear of the COVID-19	1.65 (1.50–1.80)	<0.001	1.73 (1.47–2.03)	**<0.001**
Information regarding the COVID-19 pandemic and vaccination	1.48 (1.33–1.64)	<0.001	0.98 (0.82–1.17)	0.85
Compliance with hygiene measures	1.25 (1.10–1.42)	0.001	0.95 (0.75–1.22)	0.69
Trust in COVID-19 vaccination	2.11 (1.87–2.38)	<0.001	2.11 (1.69–2.63)	**<0.001**
Fear of a second booster dose or a new COVID-19 vaccine	0.63 (0.59–0.67)	<0.001	0.66 (0.59–0.75)	**<0.001**
Concerns about the long-term side effects of a second booster dose or a new COVID-19 vaccine	0.71 (0.68–0.75)	<0.001	0.91 (0.81–1.04)	0.16
Self-perceived protection by the previous COVID-19 vaccine doses	1.29 (1.21–1.38)	<0.001	1.04 (0.92–1.18)	0.53

An odds ratio of <1 indicates a negative association, while an odds ratio of >1 indicates a positive association. Bold *p*-values indicate statistically significant associations. CI: confidence interval; OR: odds ratio. ^a^ R^2^ for the final multivariable model was 67.1%.

## Data Availability

The data presented in this study are available on request from the corresponding author.

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
