# Peer review of "Predictors of Willingness of the General Public to Receive a Second COVID-19 Booster Dose or a New COVID-19 Vaccine: A Cross-Sectional Study in Greece"

_vaccines, 2022, doi:10.3390/vaccines10071061_

Round 1
Reviewer 1 Report
Abstract:
- Line 21 to 23: Shoulde be re-worded
Introduction:
- Line 57 to 64: Please add the references
- It should be added a section on the associated factors with the willingness to accept a booster dose of COVID-19 vaccines.
Methods:
- Line 70 to 73: Please add a reference
- I suggest adding a flow-chart of study for more clearly
- Please show the method to remove the subjects who can answer the questinnaire multiple times
- Line 129 to 136: Why didn't the authors add an "other" option
- Why did the authors use the vaccination acceptance rate for the first COVID-19 booster dose rate as the reference to calculate the sample size for this study? Because the willingness rate for the first booster dose can be vastly different from the second booster's acceptance rate. (Indeed, the authors took a reference rate of 81%, but in their study, only 22.7% agreed to get the second dose of booster vaccination).
- The authors coded the outcome variable “probably yes” or “definitely yes” as willing participants to receive a second booster dose or a new COVID-19 vaccine, compared to those who answered “definitely no”, “probably no” or “unsure” as unwilling or hesitant participants to receive a second booster dose or a new COVID-19 vaccine and performed an analysis using multivariable logistic regression. This method is not ideal, in my opinion. I recommend using multinomial logistic regression
Results
- The authors calculated the required sample size to be 601, but there were 815 participants. They need a better explanation of the data collection method. In my opinion, the sample size calculation should be removed. The sample size is the total number of respondents to the questionnaire during the study period
- Predictors of willingness: please reference my comments above
Reviewer 2 Report
First of all, thank you for the opportunity to participate into the peer review of this very interesting paper on the willingness of the general Greek population to receive a second COVID-19 booster dose or a new COVID-19 vaccine.
Galanis et al. have performed their study through a consolidated and well described approach, and the results are presented clearly and through a proper approach.
In fact, they were able to identify the following facilitators towards a new vaccination:
- male gender
- self perceived physical health
- previous COVID-19 infection
- fear of COVID-19 infection
- trust in COVID-19 vaccination
- fear of the booster dose
All of the aforementioned factors are quite consistent with previous understanding of vaccine acceptance, but their identification in dealing with SARS-CoV-2 vaccine has a clear and straightforward significance.
Unfortunately, from my point of view, some improvements are required before a full acceptance of this paper:
1) from a methodological point of view, Authors should clearly describe the design of multivariable analysis: if OR were adjusted, which factors were included in the model in order to be adjusted? it is quite important, as the impact of correction was significant: see Table 3, Age (years) shifted from a POSITIVE effector (OR 1.03, 95%CI 1.02-1.04) to NEGATIVE one (aOR 0.97, 95%CI 0.95-0.99). Please amend the main text and discuss this specific shift in results.
2) the reporting of results in the main text and in the Table 3 is not consistent. For example: Fear of a 2nd dose is reported in Table 2 as aOR 0.66, 95%CI 0.59 to 0.75, while the main text (row 206-7): 0.65, 95%CI 0.58 to 0.74. Several other variables are improperly reported. Please fix.
3) Authors have discussed how the recruitment strategy may have impaired the reliability and generalizability of the results, but some further information must be provided about the parent web sites or discussion groups that were included in the analysis. Moreover, a more focused discussion on the results is required. According to your results, 2/3 of the respondents had a University level degree: even though Greece (according to OECD: https://www.oecd.org/education/education-at-a-glance/EAG2019_CN_GRC.pdf) has the the fourth highest tertiary enrolment rate among OECD countries and has experienced an increase in tertiary education attainment over the last decade, a certain oversampling of highly educated individuals is evident.
Finally, please be aware that, in the discussion section, some sentences have not been clearly amended from the template (rows 341-344)
Author Response
Dear Reviewer,
Thank you for giving us the opportunity to revise our manuscript entitled "Predictors of willingness of the general public to receive a second COVID-19 booster dose or a new COVID-19 vaccine: a cross-sectional study in Greece". We would also like to thank you for their insightful comments and suggestions on how to improve our manuscript. We respectfully tried to address the issues raised and to revise our manuscript accordingly. We hope that our revision will reach the high standards of the Journal “Vaccines”. Please note that we explain how we addressed all issues brought up in your letter and to the specific points raised by the Reviewers, in the subsequent pages of our response letter. Also, we made some changes in the manuscript according to the Editorial Office instructions.
We look forward to hearing from you
Best Regards
The authors
First of all, thank you for the opportunity to participate into the peer review of this very interesting paper on the willingness of the general Greek population to receive a second COVID-19 booster dose or a new COVID-19 vaccine.
Galanis et al. have performed their study through a consolidated and well described approach, and the results are presented clearly and through a proper approach.
In fact, they were able to identify the following facilitators towards a new vaccination:
- male gender
- self perceived physical health
- previous COVID-19 infection
- fear of COVID-19 infection
- trust in COVID-19 vaccination
- fear of the booster dose
All of the aforementioned factors are quite consistent with previous understanding of vaccine acceptance, but their identification in dealing with SARS-CoV-2 vaccine has a clear and straightforward significance.
Unfortunately, from my point of view, some improvements are required before a full acceptance of this paper:
1) from a methodological point of view, Authors should clearly describe the design of multivariable analysis: if OR were adjusted, which factors were included in the model in order to be adjusted? it is quite important, as the impact of correction was significant: see Table 3, Age (years) shifted from a POSITIVE effector (OR 1.03, 95%CI 1.02-1.04) to NEGATIVE one (aOR 0.97, 95%CI 0.95-0.99). Please amend the main text and discuss this specific shift in results.
Answer: Done
Dear Reviewer, you are right. We included all independent variables in a multivariable logistic regression model in order to eliminate confounding but we did not mention it in our manuscript. We add now this information in the methods section.
Also, we discussed the shift of the impact of age in results.
2) the reporting of results in the main text and in the Table 3 is not consistent. For example: Fear of a 2nd dose is reported in Table 2 as aOR 0.66, 95%CI 0.59 to 0.75, while the main text (row 206-7): 0.65, 95%CI 0.58 to 0.74. Several other variables are improperly reported. Please fix.
Answer: Done
Dear Reviewer, we would like to apologize for this error. We corrected the numbers in the text.
3) Authors have discussed how the recruitment strategy may have impaired the reliability and generalizability of the results, but some further information must be provided about the parent web sites or discussion groups that were included in the analysis. Moreover, a more focused discussion on the results is required. According to your results, 2/3 of the respondents had a University level degree: even though Greece (according to OECD: https://www.oecd.org/education/education-at-a-glance/EAG2019_CN_GRC.pdf) has the the fourth highest tertiary enrolment rate among OECD countries and has experienced an increase in tertiary education attainment over the last decade, a certain oversampling of highly educated individuals is evident.
Answer: Done
Dear Reviewer, we would like to apologize for the inconvenience but we cannot understand your comment “Authors have discussed how the recruitment strategy may have impaired the reliability and generalizability of the results, but some further information must be provided about the parent web sites or discussion groups that were included in the analysis” since we did not include parent web sites or discussion groups in our analysis. Probably, there is a misunderstanding.
Regarding your second comment about the oversampling of highly educated individuals in our study, we added this limitation in the Limitation section.
Finally, please be aware that, in the discussion section, some sentences have not been clearly amended from the template (rows 341-344).
Answer: Dear Reviewer, we would like to apologize for this error. We removed the rows 341-344.
Round 2
Reviewer 1 Report
Thank you
All my comments have been processed